# Efficacity of Deep Inspiration Breath Hold and Intensity-Modulated Radiotherapy in Preventing Perfusion Defect for Left Sided Breast Cancer (EDIPE): A Prospective Cohort Study Protocol

**DOI:** 10.3390/cancers15092467

**Published:** 2023-04-25

**Authors:** Jordan Eber, Cyrille Blondet, Martin Schmitt, David G. Cox, Claire Vit, Clara Le Fèvre, Delphine Antoni, Fabrice Hubele, Georges Noel

**Affiliations:** 1Department of Radiation Oncology, Institut de Cancérologie Strasbourg Europe (ICANS), 17 Rue Albert Calmette, BP 23025, 67033 Strasbourg, France; 2Department of Nuclear Medicine, Institut de Cancérologie Strasbourg Europe (ICANS), 17 Rue Albert Calmette, BP 23025, 67033 Strasbourg, France; 3Clinical Research Unit, Institut de Cancérologie Strasbourg Europe (ICANS), 17 Rue Albert Calmette, BP 23025, 67033 Strasbourg, France; 4Radiotherapy Department, ICANS, University of Strasbourg, 67098 Strasbourg, France

**Keywords:** radiotherapy, breast cancer, heart toxicity, cardiac SPECT

## Abstract

**Simple Summary:**

Radiation-induced heart disease represents a spectrum of early and late effects. It is a significant concern for cancer survivors who receive thoracic radiation therapy, as it is the most common nonmalignant cause of death in this population. Subclinical cardiac damage can be detected at an early stage after irradiation and may possibly be used to predict late cardiac complications. Modern radiotherapy techniques, such as deep inspiration breath-hold radiotherapy or intensity-modulated radiation therapy, can reduce the heart dose and probably do not cause alterations in myocardial perfusion. This paper describes the study protocol we will use to evaluate perfusion defects after left-sided breast radiotherapy using myocardial perfusion single-photon computed tomography.

**Abstract:**

Breast radiotherapy can lead to radiation-induced cardiac disease, particularly in left breast cancers. Recent studies have shown that subclinical cardiac lesions, such as myocardial perfusion deficits, may occur early after radiotherapy. The primary method for irradiating breast cancer, known as opposite tangential field radiotherapy, can cause the anterior interventricular coronary artery to receive a high dose of radiation during left breast irradiation. To explore alternative approaches that could reduce the risk of myocardial perfusion defects in patients with left breast cancer, we plan to conduct a prospective single-center study using a combination of deep inspiration breath hold radiotherapy and intensity modulated radiation therapy. The study will use stress and, if necessary, resting myocardial scintigraphy to assess myocardial perfusion. The trial aims to show that reducing the cardiac dose with these techniques can prevent the appearance of early (3-month) and medium-term (6- and 12-month) perfusion disorders.

## 1. Introduction

Breast irradiation can lead to incidental irradiation of the heart included in the irradiation field, resulting in an increased risk for a spectrum of early and late complications known as radiation-induced heart disease (RIHD) [1,2]. In 2013, Darby et al. found that there is a connection between the mean heart dose (MHD) and an increased risk of major coronary events. The relative risk of such events increased by 7.4% per gray (Gy) without any threshold following breast irradiation [2]. This finding has been confirmed in subsequent studies on breast cancer patients treated with three-dimensional conformal radiotherapy (3D-RT), which have shown increases in relative risk of ischemic heart disease of 4% and 6.4% per Gy of MHD [3,4].

However, these late effects can be considered definitive and can be corrected. One of the aims of a radiation oncologist is to avoid all risks of side effects, principally heart complications. Earlier research has established that perfusion defects are correlated with wall motion abnormalities, decreases in left ventricle ejection function, or the manifestation of clinical cardiac symptoms [5,6,7]. Lack of early event detection could be the proof of heart-sparing and the hope to eliminate the late effect described by publications.

Within a year following radiation treatment of the left breast, an elevation in cardiac uptake can be observed on 18F-fluorodeoxyglucose positron emission tomography, even in the absence of coronary stenosis. This finding suggests a potential dysfunction of microvascularization [8]. Perfusion defects may be irreversible or reversible, according to resting single photon emission computed tomography (SPECT) studies. Infarcted myocardium denotes irreversible defects, whereas ischaemic myocardium is diagnosed through reversible defects [9,10,11,12,13,14].

Cardiac SPECT is a sensitive and specific technique capable of detecting perfusion abnormalities. In several studies, it has been reported that RIHD after adjuvant radiation for left-sided breast cancer may occur earlier and can be detected by cardiac SPECT [5,10]. The rate of perfusion abnormalities observed by cardiac SPECT ranges from 27 to 70% in patients after breast irradiation [9,10,11,12]. A prospective study of 71 patients irradiated for breast cancer detected 42.9% perfusion abnormalities six months after radiation therapy for left-sided breast cancer [13].

In patients with left-sided breast cancer, perfusion abnormalities are more commonly observed, indicating that the incidence of these defects is concentrated in the radiotherapy field and is linked to the degree of cardiac exposure [13,14,15]. In a study of 114 patients who received breast cancer irradiation, volume-dependent perfusion defects were detected in approximately 40% of patients within two years after irradiation. The occurrence of new defects was found to be approximately 10% to 20% in patients with less than 5% of their left ventricle included within the irradiation fields and 50% to 60% in those with greater than 5% of their left ventricle included (*p* = 0.33 and *p* < 0.001) [5].

Breast cancer irradiation is mainly achieved by a tangentially opposed field approach. Considering cardiac anatomy, when treating a left breast, it would be expected to have a high dose to the left anterior descending coronary artery (LADA) but not to the left circumflex and right coronary arteries [14,16]. The distribution of the dose to the heart is heterogeneous [17], and cardiac tissues are differently radiosensitive [18,19]. Therefore, MHD may not be the most relevant dose parameter to assess cardiac exposure [20].

In the era of modern radiotherapy techniques, cardiac substructures could be delineated to optimize heart dosimetry and protect patients from cardiac complications [21]. The use of reference atlases and autosegmentation software relevant to these substructures would save time, quality, and reproducibility in delineation and consequently in dose calculation [22,23].

The correlation between perfusion defects and radiation dose to the myocardium implies that strategies aimed at reducing the radiation dose have the potential to mitigate myocardial damage. The distribution of the radiation dose to the heart is contingent on several factors, including the radiation technique utilized, the irradiated volumes, and the location of the radiation field [24,25]. A variety of radiotherapy techniques are being used to spare the heart [26] by limiting the dose to the heart (intensity-modulated radiotherapy (IMRT), proton therapy) [27], by maneuvers to increase the distance from the heart to the target volume (deep inspiration breath-hold (DIBH), prone position) [28], or by changing the target volume and not including all the breast glandular tissue (partial breast irradiation) [29]. However, since RIHD appears clinically most often many years after irradiation, the clinical impact of these techniques remains uncertain.

The DIBH technique can effectively displace the heart posteriorly, medially, and inferiorly away from the breast and the deep border of the tangential fields. Some investigations have recently been performed to aid in the prediction of major beneficiaries of the DIBH technique using this beam arrangement [30,31], so that non-beneficiaries may be candidates for alternative RT techniques such as IMRT [32]. There is a clear correlation between the incidental radiation of the heart in left breast irradiation and the detection of early postradiotherapy perfusion defects, which highlights the importance of implementing heart sparing techniques. Studies suggest that DIBH may be beneficial in preserving cardiac perfusion, particularly with low cardiac doses of less than 5 Gy at 6 months and 1 year post-radiation therapy [33,34].

Advanced radiotherapy techniques, such as IMRT and VMAT, modify the distribution of doses to the heart by increasing low doses [35,36]. Biophysical models for normal tissue complications suggest that the relationship between radiation dose and heart complications follows a sigmoidal dose-response curve in this case [36], rather than a linear curve. Therefore, the risk of heart complications, which was estimated at 7.4% per Gy by Darby et al. [2], may be overestimated when smaller, more homogeneous doses are used.

This single-institution prospective study will assess the utility of DIBH utilizing a controlled surface monitoring technique (AlignRT, Vision RT Ltd., London, UK) and IMRT as a means of preventing cardiac perfusion defects as determined by cardiac gated-SPECT in patients receiving irradiation for left-sided breast cancer.

## 2. Materials and Methods

### 2.1. Study Setting

This interventional, nonrandomized, monocentric, descriptive, and prospective pilot study is being led by the University Radiotherapy Department of the Strasbourg-Europe Cancerology Institute (ICANS).

This study was approved by the Ethics Committee of Ouest VI in August 2022 and registered on clinicaltrials.gov in July 2022 (NCT05454553).

### 2.2. Participants

For a full overview of all inclusion and exclusion criteria, see Table 1.

### 2.3. Practical Conduct of the Study

#### 2.3.1. Screening Procedures and Baseline Evaluation

Eligible patients will receive detailed information about the study, including its characteristics, consequences, and constraints, through a patient information sheet and consent form, as well as through an oral explanation by the investigator. Patients who agree to participate in the study will be required to sign an informed consent form. Before initiating radiotherapy, the patients will undergo various examinations, including a chest CT scan in the treatment position to aid in 3D treatment planning and dose calculation, a chest CT angiography to identify cardiac substructures, and baseline cardiac gated-SPECT imaging.

#### 2.3.2. Treatment Phase—Radiotherapy

During the treatment phase of this study, patients will receive a standard course of breast radiotherapy administered in our department. However, there will be a difference in the treatment planning process as cardiac substructures will be delineated on simulation CT scans with the help of chest CT angiography on the Varian Eclipse treatment planning system (Varian Medical Systems), according to Feng’s atlas [22], to ensure homogeneity and reproducibility of the cardiac segmentation process. Delineated cardiac substructures will be: left atrium, left ventricle, right atrium, right ventricle, left main coronary artery, LADA, left circumflex artery, right coronary artery, pulmonary artery, superior vena cava, aortic valve, pulmonic valve, mitral valve, tricuspid valve, atrioventricular node, pericardium, ascending and descending aorta. Cardiac substructure delineation will be validated by an experienced senior radiation oncologist. Target volumes, irradiation doses, and orientation to the radiotherapy treatment machine will depend on the clinical situation, as in daily clinical practice. Dose constraints for organs at risk follow the RECORAD 2022 recommendations [37]. Planned target volumes are defined according to ESTRO guidelines [38]. Target volume coverage is: PTV = V_95%_ ≥ 95%, D_2%_ ≤ 107%, and D_max_ ≤ 110%.

#### 2.3.3. Follow-Up Phase

The follow-up will consist of cardiac gated-SPECT at 3 months, 6 months, and 12 months after the end of radiotherapy, in addition to the usual follow-up.

Before these examinations, a consultation with a nuclear medicine physician will be carried out, which will allow the collection of clinical data, including details of eventual side effects.

If a reversible perfusion defect is detected during the follow-up period, the patient will be referred to a cardiologist for exploratory coronary angiography and, if necessary, transcutaneous angioplasty.

### 2.4. Practical Implementation of Cardiac SPECT

Cardiac SPECT will be obtained using a dedicated CZT camera (D-SPECT, Spectrum Dynamics Medical, Caesarea, Israel). According to the study protocol, patients will be scheduled to undergo cardiac SPECT before and at 3, 6, and 12 months post-irradiation. Stress electrocardiogram (ECG)-gated SPECT will be performed after infusion of 3 MBq/kg 99mTc-tetrofosmin (Myoview^®^, General Electric Healthcare, Chicago, IL, USA) at peak pharmacological stress with regadenoson (single dosage: 400 µg; Rapiscan^®^, GE Healthcare, Chicago, IL, USA). Intake of xanthic bases such as caffeine will be discontinued 24 h before cardiac SPECT. Rest ECG-gated SPECT will be performed on the same day, 4 h after stress ECG-gated SPECT, with administration of 8 MBq/kg 99mTc-tetrofosmin only if stress ECG-gated SPECT shows abnormalities.

#### 2.4.1. Cardiac SPECT Settings

The gamma camera will be configured to use a 140 keV photopeak with a 10% energy window. The acquisition time will be determined by a precount value of one hundred six counts originating from the left ventricle. The images will be reconstructed with Spectrum Dynamics software, without attenuation correction, and gated into 16 frames per cardiac cycle.

Cardiac SPECT images will provide objective quantitative data on regional myocardial perfusion, regional wall motion, and ejection fraction. The quantitative analysis will be performed using Cedars-Sinai software (New York, NY, USA) for quantitative gated SPECT (QGS)/quantitative perfusion SPECT (QPS).

#### 2.4.2. Use of Regadenoson (RAPISCAN^®^)

The quality of the cardiac SPECT results is closely tied to the quality of the stress test that is performed. The stress test can be conducted through physical exercise, such as running on a treadmill or pedaling on a bicycle. To ensure that the test is satisfactory, the patient must achieve an appropriate heart rate level of at least 85% of their maximum theoretical rate during the physical exercise stress test. However, depending on their clinical history, up to 40% to 50% of patients may not be able to reach this heart rate threshold. If a physical stress test is inadequate, not recommended, or unfeasible for the patient, administration of a pharmacological stress agent through intravenous injection can be used to induce myocardial perfusion abnormalities.

To have a secure and reproducible protocol for patients, a myocardial stress scan will be performed with regadenoson, a pharmacological stress agent for radionuclide myocardial perfusion imaging.

Regadenoson causes a rapid increase in intravascular adenosine. Less than 60 s are needed to administer regadenoson (400 µg) and 99mTc-tetrofosmin. As a result, it is expected that the heart rate will increase while the blood pressure will decrease after the injection. Patients are advised to maintain a seated or lying position and should be closely monitored at regular intervals until ECG parameters, heart rate, and blood pressure have returned to their pre-injection levels.

The most reported adverse events following administration of regadenoson as a stress agent were dyspnea (28%), headaches (26%), rash (16%), and chest discomfort (13%), which typically occurred within 30 min. Overall, regadenoson was found to be well tolerated, with an incidence of serious adverse events of 1% [39,40]. Aminophylline may be used to attenuate severe and/or persistent adverse reactions to regadenoson. Due to the rupture issues that are currently encountered with regadenoson, two other molecules should be chosen for use in patients undergoing stress ECG-gated SPECT for the purposes of this study, namely, dipyridamole or dobutamine, according to the patient’s comorbidities. In the absence of rupture, administration of regadenoson will be preferred.

### 2.5. Objectives

#### 2.5.1. Primary Endpoint

The primary outcome measure of this study is the occurrence of perfusion defects on cardiac SPECT scans during follow-up periods of 3, 6, and 12 months after radiation therapy.

#### 2.5.2. Secondary Endpoints

The secondary endpoints are as follows:-The incidence of left ventricular wall motion disorder and LVEF quantification on follow-up cardiac SPECT scans at 3-, 6-, and 12-months post-irradiation.-Measurement of the delivered doses to the cardiac volumes and their substructures.-The influence of cardiac risk factors on postradiation myocardial perfusion.-Assessment of the influence of chemotherapy/trastuzumab/trastuzumab emtansine exposure on postradiation myocardial perfusion.-Assessment of tumor-bed boost location on cardiac dose.

### 2.6. Participant Timeline

Figure 1 shows the different stages of the clinical trial.

### 2.7. Data Collection, Management, and Analysis

#### 2.7.1. Assessment of Myocardial Perfusion

A nuclear medicine physician (blinded to clinical information) will independently determine a visual score for each patient’s cardiac SPECT, aided by the quantification software (QGS/QPS Cedars-Sinai software), as performed in clinical practice. The 17-segment model cartography of the left ventricle recommended by the American Heart Association will be used for the evaluation of myocardial perfusion [41]. The relative perfusion to each segment will be quantified in five gradations of perfusion defect, with each assigned a numerical value as follows: 0 = no defect; 1 = mild defect/equivocal; 2 = moderate defect; 3 = severe defect; and 4 = absent perfusion. Normal studies, therefore, will have a summed stress score (SSS) or summed rest score (SRS) of 0 and the highest possible score of 68 (absent perfusion in all 17 segments).

#### 2.7.2. Qualitative Scoring of Changes on Cardiac SPECT

Postradiotherapy cardiac SPECT will be compared to baseline to assess changes in cardiac perfusion. Increases in SSS or SRS ≥ 3 points in one segment or ≥ 2 points in at least two segments between post- and pre-RT scans will be considered clinically significant.

#### 2.7.3. Assessment of Ventricular Ejection Function

The QGS/QPS Cedars-Sinai software will automatically calculate left ventricular ejection fractions (EFs). This will be accomplished by estimating the LV endocardial surface throughout the cardiac cycle and calculating LV volumes as the sum of the voxels within the contours of each frame. End-diastolic and end-systolic volumes will be determined from the LV volume curves, and EFs will be calculated accordingly.

#### 2.7.4. Assessment of Wall-Thickness Abnormality

The presence or absence of wall-motion abnormalities will be noted for each of the 17 cardiac segments, with any wall-thickness abnormalities visually categorized as hypokinetic, akinetic, or dyskinetic. The size of the affected wall (a small or large portion) will be classified as mild or severe.

#### 2.7.5. Dosimetric Data

Dosimetric parameters that will be collected are: mean dose (Dmean), maximum dose (Dmax), and minimum dose (Dmean) for all substructures; V5Gy and V25Gy for cardiac cavities (left atrium, left ventricle, right atrium, right ventricle); V30Gy and V40Gy for coronary arteries (left main coronary artery, left anterior descending artery, left circumflex artery, right coronary artery), according to DEGRO breast cancer expert panel constraints [42]. Data will be extracted manually from the Varian Eclipse treatment planning system (Varian Medical Systems).

#### 2.7.6. Clinical Data

Other clinical data will be collected during consultation, from clinical exams, and from medical files.

#### 2.7.7. Data Management

Data will be collected and managed with the use of the clinical data management system CleanwebTM.

### 2.8. Statistical Analysis

#### 2.8.1. Sample Size

The estimation of the anticipated incidence of perfusion defects is approximately 17%. The two radiotherapy techniques evaluated (DBIH and IMRT) reduce heart exposure to irradiation. The objective of this dose reduction is to reduce the incidence of perfusion defects to 0%. Assuming an α of 0.05 and 80% power, and since two irradiation techniques are evaluated, the study should include at least 58 subjects to ensure adequate power and compensation for possible loss of patients during the protocol (Rosner B. Fundamentals of Biostatistics, 7th ed., Boston, MA: Brooks/Cole).

#### 2.8.2. Statistical Methods

Medians, proportions, and percentages will be used to describe the population and resulting observations.

The incidence of perfusion defects on follow-up cardiac SPECT will be calculated based on the number of patients with an increase in SSS or SRS ≥ 3 points in one segment or ≥2 points in at least two segments between post- and pre-radiotherapy cardiac SPECT. For comparison among cardiac SPECT results at different time points, the Wilcoxon rank-sum test will be used. The Bonferroni technique will be applied to account for multiple comparisons. Spearman rank correlation will be employed to examine the link between cardiac SPECT changes at 6 months and irradiated heart structures. A univariate linear regression analysis will be conducted to investigate the association between various cardiovascular risk factors and alterations in cardiac SPECT defects. Standard proportional hazard regression analysis will be used for the subsequent multivariate analysis. A *p* value of 0.05 will be considered indicative of statistical significance.

#### 2.8.3. Project Duration and Expected Outcomes

The estimated period of inclusion will be one year. The estimated study completion date will be the end of 2023. The follow-up duration will be one year. The complete duration is estimated to be two years.

The study is expected to demonstrate that both radiotherapy techniques studied avoid the appearance of early- and medium-term perfusion defects. For this purpose, this trial proposes to reinforce the methodology of the previous studies by including a significant number of patients and by performing cardiac SPECT that follows a reversible abnormality detection protocol (resting and stress phases) and is administered at a frequency that covers the other data in the literature.

## 3. Discussion

Breast cancer patients are expected to have a prolonged lifespan, and the potential for a significant cardiovascular event is a significant public health concern. Recent findings indicate that RIHD may arise as an earlier complication and that subclinical cardiac damage can be detected. The conventional irradiation technique for breast cancer is opposite tangential field radiotherapy; however, this technique can cause early perfusional defects, according to the literature. Modern radiotherapy techniques, such as IMRT or DBIH, can result in reduced heart doses and probably do not cause alterations in myocardial perfusion.

## 4. Conclusions

If the hypothesis of the absence of perfusion defects is confirmed, then the benefit of the use of these two irradiation techniques will be real and could be considered as new references.

Cardiac follow-up is regularly questioned after breast radiotherapy; if these techniques show no negative impact, specific post-irradiation monitoring will probably no longer be necessary.

## 5. Patents

Declaration: Ethics approval and consent to participate.

This prospective study was approved by an Ethics Committee (the Committee for the Protection of Persons, CPP) and a competent health authority: the Agence nationale de sécurité du médicament et des produits de santé (ANSM).

## Figures and Tables

**Figure 1 cancers-15-02467-f001:**
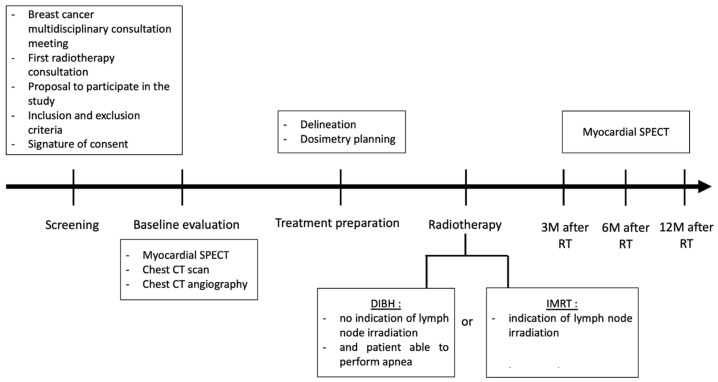
EDIPE trial schema. Abbreviations: CT: computed tomography; DIBH: deep inspiration breath hold; IMRT: intensity modulated radiation therapy; M: month; RT: radiotherapy; SPECT: single photon emission computed tomography.

**Table 1 cancers-15-02467-t001:** Inclusion and exclusion criteria.

Inclusion Criteria	Exclusion Criteria
Patients with left-sided breast cancer histologically confirmed after lumpectomy or mastectomy with/without lymph node involvement who are planned for DIBH-RT or IMRT Age > 18 years Karnofsky Performance Status > 60% Absence of psychiatric illness hindering follow-up Patient understanding French Signature of informed consent Patient registered with social insurance	Bilateral breast cancer History of thoracic irradiation Pregnancy or breastfeeding Any medical contraindication to cardiac SPECT or chest CT angiography. Any medical contraindications about regadenoson administration Patient under guardianship

Abbreviations: CT: computed tomography; DBIH-RT: deep inspiration breath hold radiotherapy; IMRT: intensity modulated radiotherapy.

## Data Availability

Not applicable.

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
