# Peer review of "Efficacity of Deep Inspiration Breath Hold and Intensity-Modulated Radiotherapy in Preventing Perfusion Defect for Left Sided Breast Cancer (EDIPE): A Prospective Cohort Study Protocol"

_cancers, 2023, doi:10.3390/cancers15092467_

Round 1

Reviewer 1 Report

Thank you for providing me the opportunity to review this paper. The manuscript is generally well-written and I read it with great interest. I would like to offer some comments as follows:

 Q1.  2.3.2 Treatment Phase – Radiotherapy

It is recommended that you describe the specific organs of the heart to be depicted and how they are contoured.

Q2.  2.4.1 Practical implementation of cardiac SPECT

Why did you select 99mTc-tetrofosmin?

Why did you not select 201TlCl or 99mTc-MIBI?

Q3. Discussion

Is it likely that the patients in the present study will prove to have more reduced impaired perfusion than those who did not undergo previous DIBH or IMRT?

Q4.

The results of the present study examine myocardial return failure over a short period of 2 years.

However, the INTRODUCTION has mixed results for long-term cardiac impairment of 5 and 10 years and should be sorted out.

Q5.

It is better to use a unified terminology for cardiac SPECT.

Author Response

Comment 1

Q1.  2.3.2 Treatment Phase – Radiotherapy

It is recommended that you describe the specific organs of the heart to be depicted and how they are contoured.

Reply

Thank you to the reviewer for pointing out the lack of detail in the contouring process of the cardiac sub-structures. As suggested by the reviewer the paragraph "2.3.2 Treatment Phase – Radiotherapy " has been modified with the addition of a precise description of the delineated cardiac sub-structures and how they will be delineated.

Modifications added to the manuscript

The text has been amended Page 5 line 164-173. Reference has been added (22).

Comment 2

Q2.  2.4.1 Practical implementation of cardiac SPECT

Why did you select 99mTc-tetrofosmin?

Why did you not select 201TlCl or 99mTc-MIBI?

Reply

We thank the reviewer for this pertinent question. Although 201Tl is one of the best tracers because of its analogy with K+, the radiation delivered to the patient is not compatible with "as low as possible" radiation protection rules. Therefore 99mTc-labelled tracers are preferred. Among the available myocardial tracers, only 99mTc-tetrofosmin requires room temperature preparation conditions, while offering similar performance as 99mTc-MIBI in the detection of myocardial perfusion.

Modifications added to the manuscript

No modification of the manuscript required.

Comment 3

Q3. Discussion

Is it likely that the patients in the present study will prove to have more reduced impaired perfusion than those who did not undergo previous DIBH or IMRT?

Reply

Abnormalities in perfusion are more frequently detected in left-sided breast cancer patients, suggesting that the incidence of perfusion defects is located in the radiotherapy field and correlated with the degree of cardiac exposure

As a result, measures to decrease the radiation dose could prevent damage to the myocardium, including early perfusion defects. Deep inspiration breath hold radiotherapy and intensity modulated radiotherapy are effective techniques for reducing cardiac doses.

Although the absence of a perfusion deficit is likely, this has not been demonstrated with a robust scintigraphy protocol such as the one proposed in our study.

Modifications added to the manuscript

No modification of the manuscript required.

Comment 4

The results of the present study examine myocardial return failure over a short period of 2 years.

However, the INTRODUCTION has mixed results for long-term cardiac impairment of 5 and 10 years and should be sorted out.

Reply

The reviewer makes an important point about the lack of explanation between late cardiac side effects and early detection of possible perfusion defect. We have reworded the introduction to make it clearer.

Modifications added to the manuscript

The text has been amended Page 2 line 54-60. References have been added (5-7)

Comment 5

It is better to use a unified terminology for cardiac SPECT.

Reply

In accordance with the reviewer's comment the text has been modified with unified terminology for cardiac SPECT.

Modifications added to the manuscript

The text has been amended Page 2 line 67, 69 and 70; Page 5 line 159, 179, 188, 190, 195; Page 6 line 202; Page 6 line 236, 241; Page 7 line 256, 265 and 266; Page 8 line 307, 309 and 310; Page 9 line 313, 315 and 325.

Reviewer 2 Report

Interesting project. 

Some methodological aspects should be described in more detail:

- Treatment phase (section 2.3.2):

Structure delineation: specific cardiac substructures should be enumerated and countouring guidelines should be refer to.

Dose constraints and treatment optimization parameters: a reference with the treatment planning methods used in the department should be included. If the reference is not available, dose constraints in organs at risk and optimization parameters for target volumes should be specified.

- Data collection (section 2.7): a subsection dedicated to dosimetric data should be included, describing in detail which dosimetric parameters will be collected and how the data will be obtained from the Treatment Planning System.

- Assesment of myocardial perfusion: the cardiac segments to be analyzed with SPECT should be cited.

Author Response

Comment

Structure delineation: specific cardiac substructures should be enumerated and countouring guidelines should be refer to.

Reply

Thank you to the reviewer for pointing out the lack of detail in the contouring process of the cardiac sub-structures. As suggested by the reviewer the paragraph "2.3.2 Treatment Phase – Radiotherapy " has been modified with the addition of a precise description of the delineated cardiac sub-structures and how they will be delineated.

Modifications added to the manuscript

The text has been amended Page 5 line 164-173. Reference has been added (22).

Comment

Dose constraints and treatment optimization parameters: a reference with the treatment planning methods used in the department should be included. If the reference is not available, dose constraints in organs at risk and optimization parameters for target volumes should be specified.

Reply

Thank you for your constructive feedback. Treatment planning methods used in the department follow the ICRU reports 50 and 62 for 3D breast treatment planning and 83 for IMRT planning.

The full treatment planning protocol has not been added to the manuscript due to its length. If the reviewer feels that it should be included in the manuscript, we can add it in Appendix with the agreement of the editor.

We have revised the manuscript to include dose constraints in organs at risk and optimization parameters for target volumes.

Modifications added to the manuscript

The text has been amended Page 5 line 175-177. References have been added (37-38)

Comment

Data collection (section 2.7): a subsection dedicated to dosimetric data should be included, describing in detail which dosimetric parameters will be collected and how the data will be obtained from the Treatment Planning System.

Reply

Thank you for your insightful review of the data collection. We appreciate your suggestion regarding the inclusion of a subsection dedicated to dosimetric data. In response to the reviewer's comment, this section was added to the manuscript.

Modifications added to the manuscript

The text has been amended Page 8 line 281-288. Reference has been added (42).

Comment

Assessment of myocardial perfusion: the cardiac segments to be analyzed with SPECT should be cited.

Reply

In response to the reviewer's comment regarding the assessment of myocardial perfusion, we have described more clearly the use of the recommended cardiac 17-segment model to assess myocardial perfusion. We hope that this updated information will enhance the clarity of our research plan.

Modifications added to the manuscript

The text has been amended Page 7 line 257-259. Reference has been added (42).

Reviewer 3 Report

Dear authors,

I want to congratulate you for having prepared such a fundamental study, which, in my opinion, has been awaited by the involved research community for a long time since actually, no evidence about the magnitude of heart-sparing effects from the use of techniques like DIBH and IMRT for left-sided breast cancer patients is available. This study will clarify if and how much our (I'm a radiation oncologist working with the DIBH technique every day) technical daily efforts correspond to a real benefit for patients. The study is well-structured and will provide the necessary answers. I have no particular comments. Only a minor one to enrich the reference list with some pertinent literature:

- at line 103, after "...away from the deep border of the tangential fields.", you could add "Some investigations have recently been done to aid the prediction of major beneficiaries of the DIBH technique using this beam arrangement [a, b], so that non-beneficiaries may be candidates for alternative RT techniques such as IMRT [c]." and cite PMID: 33788746 and PMID: 35884538 in [a, b] and PMID: 34988301 in [c].

Author Response

We warmly thank the reviewer for his comment. As he pointed out, there is a lack of evidence regarding the heart-sparing effects of techniques like DIBH and IMRT for left-sided breast cancer patients. We believe that our study will help to fill this gap and provide much-needed clarity on the benefits of these techniques for patients.

We are grateful for the support as a radiation oncologist who works with the DIBH technique regularly. The reviewer feedback and insights are valuable to us, and we hope that our study will further inform the reviewer daily efforts and decision-making for the benefit of his patients. We look forward to sharing the results of our study.

The reviewer's suggestion on text and the addition of references has been made.

Modifications added to the manuscript

The text has been amended Page 3 line 109-112. References have been added (30-32).